# The Effect of Elasticity of Gelatin Nanoparticles on the Interaction with Macrophages

**DOI:** 10.3390/pharmaceutics15010199

**Published:** 2023-01-06

**Authors:** Metin Yildirim, Agnes-Valencia Weiss, Marc Schneider

**Affiliations:** 1Department of Pharmacy, Biopharmaceutics and Pharmaceutical Technology, Saarland University, 66123 Saarbrücken, Germany; 2Department of Pharmacy Services, Vocational School of Health Services, Tarsus University, 33400 Mersin, Turkey

**Keywords:** nanotechnology, drug delivery, atomic force microscopy, young’s modulus, phagocytosis, cell uptake

## Abstract

Gelatin is a biocompatible, biodegradable, cheap, and nontoxic material, which is already used for pharmaceutical applications. Nanoparticles from gelatin (GNPs) are considered a promising delivery system for hydrophilic and macromolecular drugs. Mechanical properties of particles are recognized as an important parameter affecting drug carrier interaction with biological systems. GNPs offer the preparation of particles with different stiffness. GNPs were loaded with Fluorescein isothiocyanate-labeled 150 kDa dextran (FITC-dextran) yielding also different elastic properties. GNPs were visualized using atomic force microscopy (AFM), and force–distance curves from the center of the particles were evaluated for Young’s modulus calculation. The prepared GNPs have Young’s moduli from 4.12 MPa for soft to 9.8 MPa for stiff particles. Furthermore, cytokine release (IL-6 and TNF-α), cell viability, and cell uptake were determined on macrophage cell lines from mouse (RAW 264.7) and human (dTHP-1 cells, differentiated human monocytic THP-1 cells) origin for soft and stiff GNPs. Both particle types showed good cell compatibility and did not induce IL-6 and TNF-α release from RAW 264.7 and dTHP-1 cells. Stiffer GNPs were internalized into cells faster and to a larger extent.

## 1. Introduction

Gelatin is a natural polypeptide obtained by acidic or alkaline hydrolysis or enzymatic degradation of collagen [1,2,3]. It is classified as “Generally Recognized as Safe (GRAS)” material by the United States Food and Drug Administration (FDA) [4], and it is already used in different approved pharmaceutical, dental and regenerative medicine applications including cancer therapy, tissue scaffolds and drug delivery [5]. Gelatin has a high number of functional groups and thus the resulting particles can be easily modified for targeted drug therapy [6] opening up improved applications. Gelatin nanoparticles (GNPs) are promising drug carriers due to the polymers’ biocompatibility, biodegradability, recyclability, low cost, wide variety of sources, good surface properties, chemical modification potential, crosslinking possibility and low immunogenicity [7]. Besides this, the elasticity of GNPs can be adjusted [8]. The hydrophilic nature of GNPs makes it especially suitable for delivery of genetic material, peptides, and proteins [9,10]. GNPs have been prepared by using different methods such as nanoprecipitation, microemulsion, solvent evaporation, simple coacervation or two-step desolvation [11,12,13,14]. The colloidal properties of size, surface charge, and elasticity of GNPs are important factors influencing biological read-outs, such as cell-nanoparticle interactions [6].

Nowadays, research highlighting particle elasticity is gaining interest as promising biological behavior is connected. This was demonstrated in studies investigating tumor penetration, blood circulation time, penetration into skin or the interaction with mucus [15,16,17,18,19]. The particle elasticity is a design parameter potentially enhancing the efficiency of targeted drug delivery. Gelatin particle elasticity can be regulated by the extent of crosslinking time. However, the number of investigations is still low compared to the other abovementioned colloidal properties [8,20,21].

Macrophages are a key cell type being involved in certain inflammatory activities and the removal of foreign material. Thus, the specific knowledge about the interaction of these cells with potentially applied drug carriers is of outmost importance. This might be essential to avoid phagocytosis and respective elimination of the particles, but could also be used to address those cells for modulation of secretory activity. RAW 264.7 cell line is a popular macrophage cell line of murine origin isolated from BALB/c mice [22]. To determine signaling pathways, drug transport, mechanisms, and function of macrophages, RAW 264.7 macrophages (mouse) are suited due to long-term storage, the proliferation rate of cell lines being higher than primary cells. Thus, possible variation in macrophage phenotypes can be reduced. However, as the carrier system should be finally designed to work in humans, the use of cells of human origin is favored. A convenient cell line is the THP-1 cells differentiated from monocytes into a macrophage-like type [23,24]. With respect to particulate uptake, Minchin et al. showed similar uptake pathways for particles with a size of 220 and 250 nm of approximately 60 to 70% mediated by the scavenger receptor A in both dTHP-1 and RAW 264.7 cells [25]. However, differences occur in other uptake-related proteins and membrane composition. Caveolin-1, for example, is expressed in dTHP-1 cells but not in RAW 264.7 [26]. To address this, particle internalization as well as inflammatory response were studied in both systems.

Beningo et al. demonstrated that rigid particles were preferentially engulfed by macrophages [27]. Weiss et al. showed that the Young’s modulus of gelatin NPs correlated with cell uptake rate of pulmonary epithelial cancer cells (A549). Harder GNPs were taken up to a greater amount than softer GNPs by A549 cells [6]. With respect to the study of Anselmo et al. [18] looking at the uptake of PEG-based hydrogel NPs (overall soft) and silica nanocapsules [28] (overall hard particles), there is a range of stiffnesses not covered for which gelatin-based particles can be used, representing medium mechanic properties obtained with a natural polymer.

Besides the cellular internalization, physical and chemical properties of nanoparticles, such as size, shape, stiffness, and surface modification, can regulate activation of immune cells and animal models [29]. Inflammation is generally defined as biological response of the immune system to pathogens, toxic materials, damaged cells, or chemical irritation [30]. Cytokines are small signaling proteins secreted by various cells and have a specific role in the communication between cells. They affect pro-inflammatory and anti-inflammatory mechanisms [31]. Typically, a set of cytokines such as IL-6 and TNF-α are investigated to obtain a first impression of the cellular response on a certain condition (here presence and/or uptake of gelatin nanoparticles). IL-6 is a pro-inflammatory cytokine produced by a large variety of cells including monocytes, endothelial cells, and adipose tissue [32]. Physiologically, it is a crucial component in immune response that has a role for the response to infections and tissue injuries [33]. TNF-α is a pleiotropic cytokine produced by T-lymphocytes, natural killer cells, and activated macrophages during trauma and infection. It is a paracrine and endocrine mediator of inflammatory processes [34]. Excessive production of inflammatory cytokines is detrimental to human health, resulting in tissue damage, hemodynamic changes, and organ failure [30]. Therefore, the main target is to design NPs that are safe, immunologically inert, and biodegradable [35].

In the present manuscript, we investigated for the first time the effect of gelatin nanoparticles with varying elasticity on macrophage uptake and cytokine secretion. To be safe, drug delivery vehicles like GNPs should not be cytotoxic and immunogenic even if their physical properties are changed. To better understand the potentially immunomodulatory or cytotoxic effects of GNPs, cytotoxicity and immunoassays of GNPs with different stiffness need to be evaluated. Thus, the effect of GNPs with different Young’s moduli on cell viability, cytokine release, and cell uptake in human and murine macrophage cells in vitro were investigated.

## 2. Materials and Methods

### 2.1. Materials

Gelatin B (average Mw 20 to 25 kDa, bloom strength 75 g), Poloxamer 188, 25% aqueous glutaraldehyde solution, branched polyethyleneimine 25 kDa (bPEI) and solvents have been purchased from Sigma Aldrich (Steinheim, Germany). Sodium metabisulfite was purchased from Merck (Darmstadt, Germany). Fluorescein isothiocyanate-dextran 150 kDa (FITC-dextran) was derived from TdB (Uppsala, Sweden). RPMI-1640 cell culture medium, HBSS buffer and 4′,6-Diamidino-2-phenylindol solution (DAPI) were obtained from Sigma Aldrich LifeScience GmbH (Seelze, Germany). Fetal calf serum (FCS) was purchased from Lonza, (Basel, Switzerland) and glutamine from Gibco–Thermo Fisher Scientific (Darmstadt, Germany). IbiTreat^®^ microscopy chambers were derived from Ibidi GmbH (Martinsried, Germany). Wheat Germ Agglutinin Alexa Fluor™ 633 conjugate (WGA Alexa Fluor™ 633) was purchased from Thermo Fisher Scientific Inc. (Darmstadt, Germany). Mouse and Human Interleukin 6 (IL-6) as well as Mouse and Human Tumor Necrosis Factor Alpha (TNF-α) ELISA kits were purchased from Shanghai Korain Biotech (Shanghai, China).

### 2.2. Preparation of Gelatin Nanoparticles

Nanoprecipitation was used to prepare gelatin nanoparticles [2]. The method was slightly adapted by the addition of sodium metabisulfite to stop the crosslinking reaction. This approach enabled the tuning of the elasticity [8]. In brief, 20 mg of gelatin B was dissolved in 1 mL MilliQ^®^ water at 50 °C under continuous stirring. The gelatin solution was injected into the antisolvent containing 450 mg poloxamer 188 (in 15 mL acetone and 1 mL water) with an injection rate of 250 mL/min with a syringe pump (Legato 200 from KD Scientific, Holliston, MA, USA) using 5 mL syringes and 0.55 × 55 mm needles. The obtained GNPs were crosslinked with 500 μL of a 1.85% glutaraldehyde solution in acetone for 1 and 3 h to obtain soft and hard GNPs, respectively. Sodium metabisulfite aqueous solution (600 mg/5 mL) was used to inactivate glutaraldehyde, and therefore terminate crosslinking. Then, GNPs were washed 3 times with water by centrifugation at 700× *g* for 8–10 min at 15 °C. Then, 150 kDa FITC-dextran-loaded GNPs were prepared by the same method with the difference that gelatin was only dissolved in 800 μL MilliQ^®^ water. Before the precipitation step, 200 µL of an aqueous 5 mg/mL FITC-dextran 150 kDa solution was added.

### 2.3. Determination of Particles’ Hydrodynamic Diameter

The particle sizes and the particle size distribution in terms of the polydispersity index (PDI) of GNPs were measured by dynamic light scattering (DLS) with a Zetaziser Ultra (Malvern Panalytical, Malvern, UK). After the last purification step, samples have been diluted 1:20 in HBSS buffer and a volume of 1 mL was investigated. Each sample was measured using square polystyrene cuvettes (DTS0012, Malvern, UK) at 25 °C in a technical triplicate, which consisted of 12 runs per measurement, as well as an experimental triplicate.

### 2.4. AFM Measurements

Visualization as well as determination of mechanical properties (Young’s moduli) were achieved by atomic force microscopy with a JPK NanoWizard^®^ III AFM (JPK Instruments, Berlin, Germany) using a commercial quadratic pyramidal tip (MLCT type, Cantilever C) purchased from Bruker Nano Inc. (Wissembourg, France). Cantilevers were calibrated prior to each measurement to determine the actual sensitivity and spring constant. Measurements were performed one day after particle preparation under liquid conditions. Therefore, the quantitative imaging mode (QI™) was applied in MilliQ^®^ water at 37 °C. Particles were electrostatically fixed to freshly cleaned and branched polyethylenimine (bPEI, 25 kDa)-coated ultraflat SiO_2_ substrates on the day of the measurement. Glutaraldehyde crosslinked GNPs are negatively charged with a surface potential of around −20 mV at neutral pH, whereas bPEI-coated surfaces have a positive surface charge. Measurement settings were fixed to an image size of 5 × 5 µm with 125 × 125 pixels with a pixel time of 50 ms, a z-length of 700 nm and an applied force of 1 nN. For data evaluation, the JPKSPM Data Processing program was used. Pixels representing the middle of a particle were manually selected and the respective force–distance curves were treated to be able to fit the corresponding Young’s moduli according to common protocols. In brief, cantilever calibration was applied, and the data were smoothed followed by baseline subtraction and contact point determination before the vertical tip position correction was applied. Subsequently, the Young’s modulus was extracted fitting the Hertz/Sneddon model with corrections for quadratic pyramidal tips according to Bilodeau [36] and assuming a half angle of 22° and a Poisson’s ratio of 0.5. Young’s moduli were obtained from 1 h and 3 h crosslinked FITC–dextran-loaded GNPs from 3 independently prepared batches. For each batch, at least 120 force–distance curves were evaluated.

### 2.5. Routine Cell Culture

Mouse macrophage cell line RAW 264.7 (TIB-71™) and the human acute leukemia monocytic cells THP-1 (ATCC-88081201) were purchased from ATCC (American Type Culture Collection, Rockville, MD, USA). The cells were cultured in RPMI-1640 supplemented with 10% heat-inactivated fetal calf serum at 37 °C in a humidified 5% CO_2_-containing atmosphere. THP-1 cells were differentiated (to dTHP-1) with 100 nM phorbol-12-myristate-13-acetate (PMA; Sigma Aldrich, Taufkirchen, Germany) for 72 h prior to each experiment and then the medium was exchanged with fresh medium without PMA. Passages between 10 and 30 were used for in vitro experiments.

### 2.6. Cytotoxicity Analysis

MTT assay was used to evaluate cytotoxic effects of GNPs on RAW 264.7 and dTHP-1 cells. RAW 264.7 cells were seeded in 96-well plates (Greiner Bio-one, Frickenhausen, Germany, 50 × 10^3^ cells/well in a medium volume of 200 µL) and grown for 48 h to allow to reach confluence. THP-1 cells were seeded in 96-well plates (80 × 10^3^ cells/well in 200 µL) supplemented with PMA (30 ng/mL) and cultured for 72 h to achieve differentiation. When confluence was achieved, cells were washed with HBSS and subsequently 150 kDa FITC–dextran-loaded GNPs in concentrations from 5 to 1000 µg/mL in HBSS were added to the cells and incubated for 24 h at 37 °C in 5% CO_2_-containing atmosphere. After the incubation time, particle suspensions were removed, and cells were washed with fresh HBSS. Cells were incubated with 10% MTT reagent (diluted in HBSS) for 4 h at 37 °C. The medium was discarded and 100 μL dimethyl sulfoxide (DMSO) was added. Subsequently, the plate was shaken and kept in the dark at room temperature for 15 min. Positive and negative control groups were treated with medium and 2% Triton-X, respectively. The absorbance was measured by using a microplate reader (TECAN infinite m200, Männedorf, Switzerland) at a wavelength of 550 nm [37].

### 2.7. Cellular Uptake of Gelatin Nanoparticles

To investigate the uptake of GNPs in RAW 264.7 and dTHP-1 cells, RAW 264.7 cells were seeded in μ Slide 8-well chambered coverslip (10^4^ cells per well). THP-1 cells were seeded (10^4^ cells per well) after being differentiated as previously mentioned. Then, the cells were incubated with 200 μL of a 1 mg/mL FITC–dextran-loaded GNP (1 and 3 h crosslinked) suspension for 4, 8 and 24 h at 37 °C. Subsequently, cells were washed twice with HBSS and fixed with 4% paraformaldehyde solution for 20 min. Nuclei were stained using 300 nM DAPI for 20 min and cell membranes were stained using 100 µg/mL WGA-Alexa Fluor^®^ 633 conjugate for 30 min. The signals of DAPI, Alexa Fluor^®^ 633 and FITC–dextran were detected after excitation at 405, 633 and 488 nm and emission at 456, 693 and 564 nm, respectively. Cell visualization for uptake evaluation was observed by confocal microscope (LSM710, Carl Zeiss AG, Oberkochen, Germany).

For quantitative analysis of the uptake, 5 × 10^4^ cells/well of RAW 264.7 and dTHP-1 cells were seeded in 24-well plates and grown for 24 h. A total of 200 µL of a 1 mg/mL FITC–dextran-loaded GNP suspension was added to the wells and incubated for 4, 8 and 24 h. After incubation, cells were washed 5 times with HBSS. Then, cells were incubated with cell lysis buffer (9803, Cell signaling technology Europe, Leiden, The Netherlands) for 30 min. The fluorescence intensity of 150 kDa FITC–dextran-loaded GNPs was measured with fluorescence microplate reader (TECAN infinite m200, Männedorf, Switzerland) [38]. The uptake efficiency as a representation of the amount of overall uptaken particles was determined with the following equation:Cellular uptake efficiency = Is − Inc/Ip − Inc × 100%, 
where Is represents the fluorescence intensity of sample after incubation with GNPs, Inc represents the fluorescence intensity of negative control group, and Ip represents the fluorescence intensity of GNPs in the concentration of 1 mg/mL. This equation was applied after assuring a linear relationship between fluorescence signal and the used particle number (Appendix A).

### 2.8. Determination of Cytokine Release after Exposure to GNP

The levels of released IL-6 and TNF-α were determined using ELISA kits according to the manufacturer’s instructions. Briefly, RAW 264.7 and dTHP-1 cells were plated at an amount of 5 × 10^5^/well in 24-well plates. After 24 h, the nonadherent cells were removed by 3 times washing with PBS. The adherent cells were then incubated for 4, 8 and 24 h with 200 µL of soft and hard GNPs in a concentration of 1 mg/mL in HBSS. Following incubation, supernatants were collected. In total, 40 µL of supernatant, 10 µL of antibody and 50 µL of streptavidin–HRP were added to precoated ELISA plate wells. Then, covered plates were incubated for 60 min at 37 °C. After removal of the reaction mixture, each well was washed five times with washing buffer. Subsequently, 50 µL of substrate solution A and 50 µL of substrate solution B were added and incubated for 10 min at 37 °C in the dark. After adding 50 μL of stop solution, the optical density of the samples was measured at a wavelength of 450 nm using the TECAN microplate reader infinite m200. Samples from untreated cells were used as negative control and cells treated with LPS (induced with 1000 and 200 ng/mL for RAW 264.7 and dTHP-1, respectively) were used as a positive control [39].

### 2.9. Statistical Analysis

Data were analyzed using GraphPad Prism version 8.0.0 for Windows (GraphPad Software, San Diego, CA, USA). Statistics analyses for comparisons of groups were evaluated by using ANOVA and unpaired Student’s *t*-test. At least 3 independently repeated measurements were used, and data were expressed as mean ± standard deviation.

## 3. Results and Discussion

### 3.1. Colloidal Properties of GNPs

The GNPs hydrodynamic diameters and particle size distribution were measured by Zetasizer Ultra in HBSS after purification. The particle size of the particles prepared with different crosslinking time was between 245 and 279 nm. The results are expressed as mean value of three batches. The average PDI values are lower than 0.23, indicating a narrow size distribution [40] (Figure 1, and size distribution graph Appendix A).

The literature for the size dependency of cellular uptake of submicron particles is not consistent. While Leclerc et al. claim particles around 250 nm to be internalized with the lowest efficiency [41], Minchin et al. showed cellular interaction for RAW 264.7 between 90 and 1000 nm and slight decrease in particle–cell interaction for particles smaller than 250 nm for dTHP-1 cells [25]. However, small particles were rather internalized, whereas the bigger particles seemed to be only associated with the cells. Therefore, the sizes of the obtained particles seem to be suitable for the subsequent cell internalization studies and no further optimization regarding the particle size is needed.

### 3.2. AFM Measurements

For imaging GNPs by AFM, the QI^TM^ mode was used in liquid at a temperature of 37 °C. AFM images showed a spherical morphology of GNPs (Figure 2). The particle size of 1 h and 3 h crosslinked GNPs was determined to be 217 ± 52 and 224 ± 65 nm, respectively. The particle size and size distribution in AFM matches with the obtained ZetaSizer results also taking the different effects on sizes into account (hydrodynamic size for DLS and the tip-broadening introduced by AFM). In addition, the measurement at liquid conditions is avoiding the collapse of the particles reflected in the height scale and heights of the particles. Young’s moduli of all measured GNPs one day after preparation were around 4.12 and 9.8 MPa, respectively. At least 15 particles were analyzed for each batch. The difference was statistically significant (Figure 3). These values prove that increased crosslinking time led to an enhancement in the stiffness of GNPs as shown before [8]. This was exactly what was intended: to only change the elasticity of the carrier system without influencing other colloidal properties.

### 3.3. Cell Viability

To detect a possible cell viability reducing effect and to exclude an influence from the particle elasticity on the cell viability, particles were tested on dTHP-1 and RAW 264.7 macrophages. The viability of dTHP-1 and RAW 264.7 cells after incubation of GNPs for 24 h was tested in terms of their mitochondrial activity by MTT assay (Figure 4). The cell viability in each group was above 90% relative to the controls for all tested concentrations of GNPs ranging from 5 to 1000 µg/mL, demonstrating that GNPs were nontoxic even at high concentrations. This indicates that the modified preparation procedure is not leading to any critical residues or changes. Furthermore, an elasticity dependent negative effect of GNPs on the metabolic activity of the tested macrophage cell lines can be excluded.

### 3.4. Cell Uptake

By using confocal microscopy, the localization of FITC–dextran-loaded GNPs inside the cells could be confirmed. Cell membranes were stained to distinguish between the nanoparticles inside the cells from those adhered to the cell surface. Most nanoparticles were scattered throughout the cell but not in the nucleus. FITC–dextran-loaded 1 and 3 h crosslinked GNPs (1 mg/mL) were incubated on RAW 264.7 and dTHP-1 cells for different incubation times (4, 8 and 24 h) before CLSM analysis. As shown in Figure 5, the cellular uptake of both GNP formulations exhibited obvious time-dependent behavior for 4, 8 and 24 h for both cell lines. However, GNPs interacted differently with RAW 264.7 and dTHP-1 cells represented by the larger extent of uptake by RAW 264.7 cells. Looking in detail into the impact of the elasticity on the particle uptake into RAW 264.7 macrophages, it can be observed that the harder particles already showed an uptake after 4 h, whereas only after 8 h an uptake for softer particles can be observed, which is further increased after 24 h. In the images obtained after 4 h of particle incubation of dTHP-1 cells, it was observed that the uptake of particles was low. However, for 8 and 24 h, it is clear that the particles are taken up in high amounts by the cells.

These data clearly demonstrate the effect of the mechanical properties of GNPs on cellular uptake. It was shown that the uptake rate by the cells depend on Young’s moduli with stiffer particles to be better internalized. This behavior is in agreement with data shown before but using different particles with higher Young’s Moduli [42]. Furthermore, the behavior is concurrent also for epithelia and other cells [6,18].

To quantify the cellular uptake of nanoparticles from the different treated groups, a microplate-based cell uptake study was performed [6,38,43]. To analyze the uptake efficiency of GNP containing 150 kDa FITC–dextran, the fluorescence intensity in the lysed cells after removal of nanoparticle suspension and thorough washing was used. In general, the level of nanoparticle uptake was lower in dTHP-1 cells than in RAW 264.7 cells through the quantitative evaluation. This result was in accordance with the qualitative data obtained from CLSM (Figure 5). This is in accordance with the literature which points to the similarity of the uptake processes for different species resulting in the same trend [44]. However, the specific numbers might depend on the cell sources (Human vs. mouse). Elasticity of hydrogel nanoparticles affects their interactions with the macrophage cells [20]. By comparing cellular uptake of soft and stiff GNPs by dTHP-1 cells, it was shown that the cell uptake efficiency in 3 h crosslinked GNPs was approximately twice as efficient and with this significantly higher than the uptake of 1 h crosslinked GNPs at each time point (*p* < 0.05) (Figure 6). The higher uptake of harder GNPs in RAW 264.7 macrophages could be observed as well but was not as pronounced as for dTHP-1 cells. This is as well reflected in the fact that significant differences for the uptake of the differently stiff particles was only determined for the first 8 h. After a 24 h incubation time, there is still a difference in uptake visible, but it was not detected to be statistically different anymore. This is overall not surprising, as one would expect that in dependence of the time and the uptake kinetics, there might be a levelling of the uptake between different particles as the slower ones can catch up [45]. Nevertheless, the overall particle uptake in RAW 264.7 is much higher than the uptake in the dTHP-1 cells. The results together indicate that stiffness of nanoparticles can affect phagocytic interaction of macrophages [46]. Soft particles can deform during membrane folding, which could reduce their uptake by macrophages [47]. Furthermore, they do not stimulate the formation of phagosomes, therefore macrophages preferred rigid ones [28,48]. Hui et al. determined that silica nanocapsules (SNCs) with larger stiffness had a higher uptake rate (three times higher) by macrophages compared to SNCs capsules with lower stiffness [28].

### 3.5. The Effect of GNPs on Cytokine Secretion

IL-6 and TNF-α are commonly used as inflammation markers [22]. Gelatin is mostly known for its pro-inflammatory activity [49]. To explore the potential immunomodulatory effect of GNPs on RAW 264.7 and dTHP-1 macrophages, the secretion of cytokines from the cells in response to GNP exposition was investigated. Besides the effect of the material and the particles, the change in Young’s Modulus was evaluated by treating the macrophages with soft and stiff GNPs at the nontoxic concentration of 1 mg/mL (see Figure 4). LPS induces the production of cytokines including IL-6 and TNF-α by macrophages [50]. Gaglio et al. have determined that PLGA nanoparticles did not induce pro-inflammatory cytokines including IL-6 and TNF-α in dendritic cells [51]. A similar relationship was obtained by Bancos et al. in their studies; gold nanoparticles had no effect on cytokine production in RAW 264.7 macrophages [52]. As shown in Figure 7A,B, very low levels of cytokines were observed in GNP-treated dTHP-1 and RAW 264.7 cells. GNP treatment did not induce IL-6 and TNF-α production at all concentrations (*p* > 0.05). The level of TNF-α and IL-6 concentrations confirmed that gelatin nanoparticles have no effect on inflammatory response in macrophage cells compared to the effect observed with LPS. Furthermore, not only the absence of an effect due to the material is detected, but also the fact that the different elasticities of the particles are not influencing IL-6 or TNF-α secretion.

## 4. Conclusions

In this study, we investigated the relationship between mechanical properties of GNPs in terms of Young’s moduli and their cellular uptake by macrophages. The potential drug delivery applications of nanoparticles require understanding of the interaction between the carriers and the tissue. This holds especially true for the investigation of the comparable new design parameter for nanoparticles: the elasticity. The adjustment of the particles’ stiffness for systematic studies is crucial, and GNPs were successfully prepared and characterized. By the variation of the crosslinking time, we were able to vary the resulting particle elasticity. This is to a great extent due to the difference in the crosslinking density. The benign character of the particles was shown on human-derived and rodent-derived macrophages by cell viability tests. In addition, for the two types of macrophages from different species, a preferred uptake of stiffer particles over softer particles is observed. Although the nanoparticles are very similar in size, the 3 h crosslinked, stiffer gelatin nanoparticles were internalized into cells faster and to a larger extent. After 24 h, stiffer GNPs were uptaken 1.37- and 2.16-fold higher by RAW 264.7 and dTHP-1 cells, respectively. The current study underlined that the elasticity is a relevant parameter regarding cellular interaction and especially uptake. For potential applications, the influence on the secretory status of the macrophages would also be of interest. For the two macrophage cell lines used, we also have shown that 150 kDa FITC–dextran-loaded soft and stiff GNPs did not trigger IL-6 and TNF-α release from macrophages. Thus, no activation of inflammatory pathways via those cytokines was initiated for a 24 h exposure using those cells. The applicability of the hydrophilic carrier system is not compromised changing the elasticity in the range that was accessible (~4–10 MPa) during our experiments. This study showed the vital role of elasticity of gelatin nanoparticles in tuning their macrophage uptake and may shed light on the design of drug carriers.

## Figures and Tables

**Figure 1 pharmaceutics-15-00199-f001:**
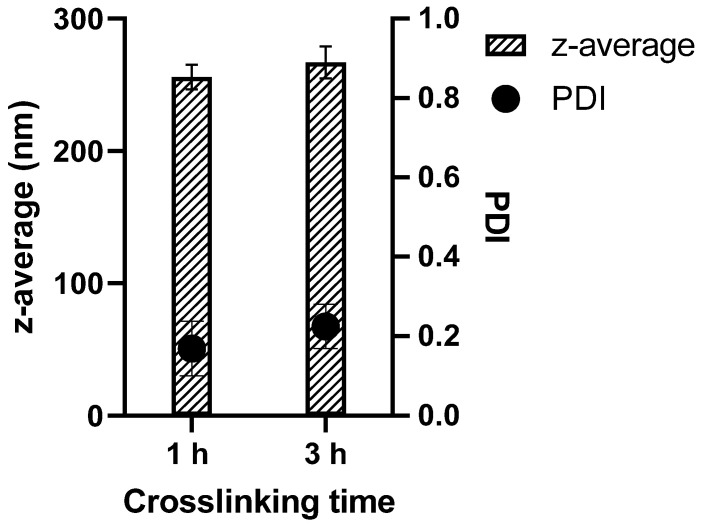
Particle sizes of soft (1 h crosslinking) and hard (3 h crosslinking) GNPs as well as the corresponding PDIs.

**Figure 2 pharmaceutics-15-00199-f002:**
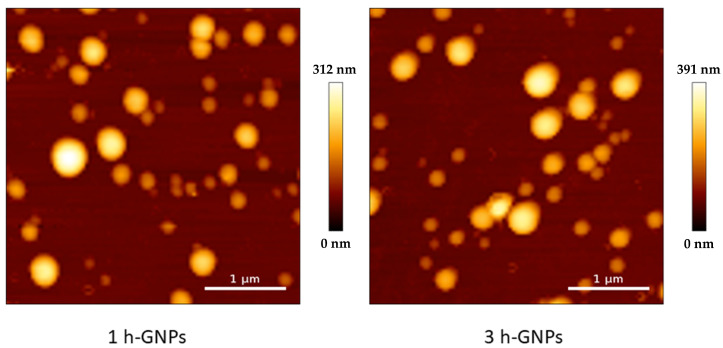
AFM height image of 1 and 3 h crosslinked GNPs measured in liquid at 37 °C, showing a number-based size of 217 ± 52 and 224 ± 65 nm.

**Figure 3 pharmaceutics-15-00199-f003:**
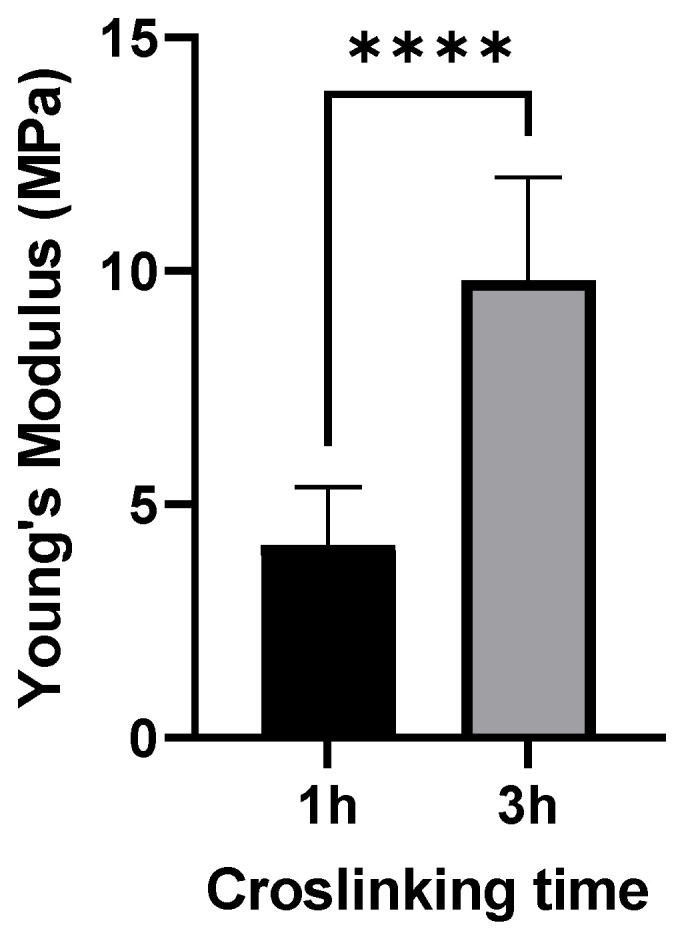
Young’s moduli for 150 kDa FITC-dextran-loaded GNPs produced with the two different crosslinking times applied in this study. Data is presented as the mean ± SD of three independently prepared batches, **** 3 h crosslinked particles are significantly different from 1 h crosslinked GNPs (*p* < 0.001).

**Figure 4 pharmaceutics-15-00199-f004:**
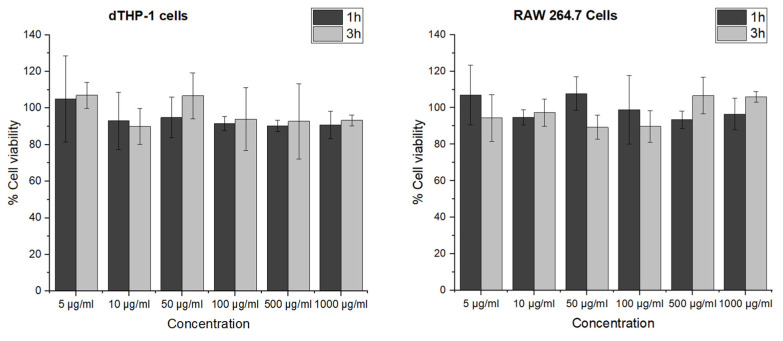
Cell viability of dTHP-1 and RAW 264.7 cells after 24 h incubation with 1 and 3 h crosslinked GNPs determined by MTT assay. The data represent the means ± SD of three independent experiments.

**Figure 5 pharmaceutics-15-00199-f005:**
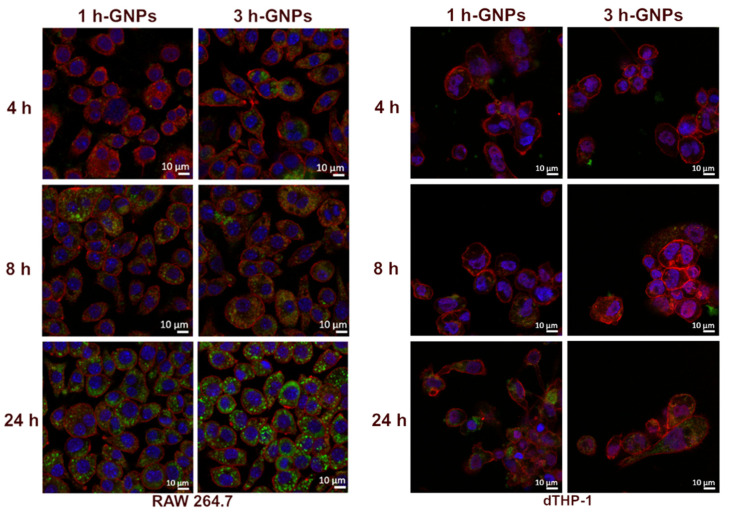
Cellular interaction of GNPs with RAW 264.7 and dTHP-1 cells, after 4, 8 and 24 h incubation of 1 (soft) and 3 h (stiff) crosslinked GNPs. The blue signal corresponds to nuclei stained with DAPI, the red signals correspond to cell membranes stained with WGA-Alexa Fluor^TM^ 633, and the green signals originate from 150 kDa FITC–dextran-loaded GNPs.

**Figure 6 pharmaceutics-15-00199-f006:**
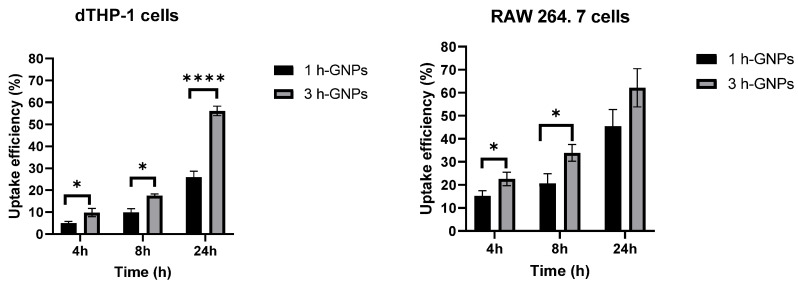
Cell uptake efficiency of GNP (1 h, soft and 3 h crosslinked, stiff) by RAW 264.7 and dTHP-1 cells for 4, 8 and 24 h. Data is presented as mean ± SD as percentage of the applied particle dose, * Different from 1 h crosslinked GNP (*p* < 0.05), **** Different from 1 h crosslinked GNP (*p* < 0.001).

**Figure 7 pharmaceutics-15-00199-f007:**
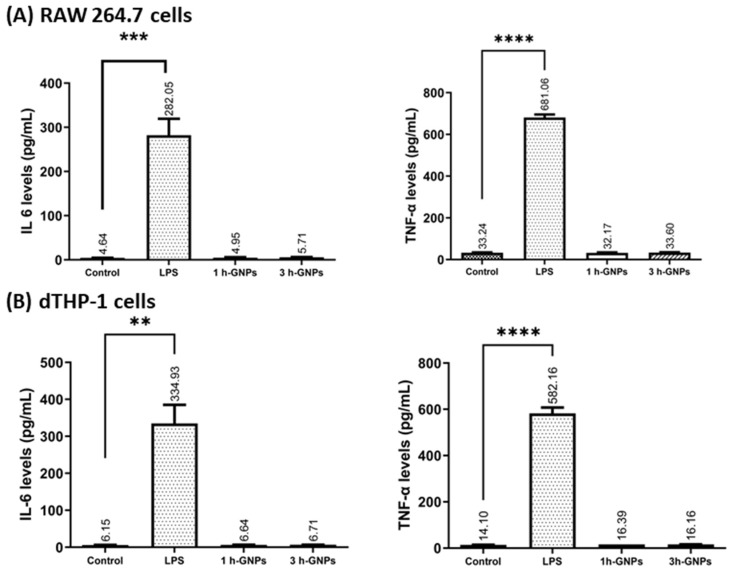
The effect of GNPs on IL-6 and TNF-α production in RAW 264.7 (**A**) and dTHP-1 (**B**) cells. Level of cytokine expression in the culture media was measured using a commercial ELISA kit. Data represent the means ± SD of three independent experiments. ** *p* = 0.0031, *** *p* = 0.0022, **** *p* < 0.001 in comparison to untreated controls; significant difference was determined using unpaired Student’s *t*-test. For positive control, RAW 264.7 and dTHP-1 cells were induced with 1000 and 200 ng/mL LPS, respectively. Untreated cells were used as negative control.

## Data Availability

The data presented in this study are available on request from the corresponding author.

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
