# Peer review of "The Effect of Elasticity of Gelatin Nanoparticles on the Interaction with Macrophages"

_pharmaceutics, 2023, doi:10.3390/pharmaceutics15010199_

Round 1

Reviewer 1 Report

Plagiarism was found 42%. Introduction need to be rewritten with more significance to the related work. More references are needed in the results and discussion. Minor language modification is needed.

Author Response

We thank reviewer 1 for the comments.             
The reviewer is right, that adding the manuscript into Turnitin an overall number of similarities of 41% is obtained. However, looking at the details it is obvious that only 3% maximum where identical in one of the sources, here an article on a similar topic from our group. 96 positions were mentioned of with 91 are below 1% and are connected to fixed expressions such as “RAW 264.7”, “macrophage cell line”, “Atomic force microscopy”, “Food and Drug Administration (FDA)” etc. just to mention a few examples. Furthermore, all the literature cited is also found as similar and included in the 42%. Looking into the introduction it is obvious that only small parts of very general aspects are similar which is not so surprising as this is the idea of the introduction to come up with such general issues. In the methods and materials part we find then the highest similar which is due to the naming of the materials and the description of similar methodological approaches. Just by eliminating the literature and those common expressions we reach below 20%. Looking into the rest in more detail clearly shows that no parts were taken from other sources.
Consequently, plagiarism is not an issue.

The introduction is rewritten especially in respect to Reviewer 2s comments, which in our opinion helped a lot to improve the introduction to motivate the performed research in a more focused manner. Furthermore, we also tried to improve the language throughout the manuscript.         
With respect to the used references in the results and discussion part we added a few more references. In total we cite nearly 50% of the references in the results and discussion part. 13 of them are exclusively used in this section. We hope we could enhance the quality of this part.

Reviewer 2 Report

Yildirim and co-workers present a novel work that correlates the elasticity of nanoparticles and their internalization by macrophages. Gelatin was chosen as the model material for this study, and both human- and animal-derived macrophages have been employed. The main conclusions point to a larger extent and a faster rate of internalization when using the stiffest nanoparticles.

The topic is highly appealing to the readers. As most studies focus on the influence of size, shape and/or surface functionalization of particles aimed at cell targeting and drug delivery, studies such as this one focusing on the mechanical properties of the systems have been scarce and are novel. The techniques that were used are in general sufficient to draw the conclusions claimed by the authors, but the presentation of the Results and Discussion can be significantly improved. I also found that the Introduction should be revised to provide a better motivation for this work.

In order to address the abovementioned concerns, I invite the authors to go through the points listed below:

1. The text should be revised. The two sentences that start in lines 41 and 42 both mention "studies"; in lines 55 and 56 the function of macrophages is mentioned, but it is repeating the idea from lines 48-52. These are just a few examples that do not help with the fluency of the manuscript.

2. The authors introduce gelatin nanoparticles (GNPs) as biocompatible, biodegradable, recyclable, low cost and low immunogenic (lines 34-35). These properties are common to several other polymer- and lipid-based NPs, so how do GNPs differ from other drug delivery systems? The advantages are not clear.

3. In lines 52-55, the text focuses on cancer treatments. Since this manuscript is not directed to a specific treatment or disease, I suggest that either this example is eliminated or other examples are added to illustrate potential applications.

4. Section 2.3 can be improved. For example, how many runs were set per measurement? What temperature was selected? What type of cuvette was used, and how much volume (important since it may affect the number of counts and quality of measurement)? While they may seem trivial, this info is aligned with the level of detail and attention given to the other methods described in Section 2.

5. In Section 2.4, it is mentioned that the GNPs were immobilized electrostatically (lines 137 and 138). The authors should state the charges of GNPs and polyethylenimine.

6. In the Results and Discussion, line 229, the average size of the GNPs is given with one decimal case. Does the equipment provide an accurate reading to this number of digits? Please consider rounding up to the units. The authors should also mention if these are mean values or the upper and lower values obtained from individual experiments.

7. I am also concerned about the claim provided in lines 231 and 232: "to only change the elasticity of the carrier system without influencing other colloidal properties". At this point only the results concerning the size have been shown, whereas the elasticity assessment only comes in Section 3.2: this claim could be made there, but not in Section 3.1. The authors should discuss if the size of the GNPs is adequate for the subsequent internalization studies (e.g. do NPs of this size usually enter cells easily?), and if it was expected that the size would affect the elasticity.

8. The distribution plots of the GNPs should also be shown, not just the z-average. This would show alongside the PDI how the crosslinking affects the size distribution.

9. The authors state in line 242 the sizes of the GNPs measured in the AFM. Again, please check the decimal cases.

10. Later it is mentioned that the AFM was made in liquid conditions, but this was not clear from the Materials and Methods. Please revise the experimental description in Section 2.4.

11. In line 249 a statistical significance is indicated (p <0.001). The p-value should be placed in the figure caption, not the Discussion.

12. The authors used dTHP-1 and RAW 264.7 cells to make the cell experiments, but the motivation for these choices is not clear. For example, dTHP-1 differentiates from the THP-1 human monocytic leukemia cell line. What are the benefits of choosing this specific cell line?

13. In line 307 differences in internalization are justified by one macrophage being human and the other from mouse. Do the authors know how they differ (e.g. is the cell surface or cell metabolism different)?

14. In Section 3.5, lines 342-346 are overlapping with the Introduction. Please consider moving this part to the Introduction and rewrite accordingly.

Author Response

Reviewer 2: We should improve the introduction, the description of the methods and the presentation of the results

Yildirim and co-workers present a novel work that correlates the elasticity of nanoparticles and their internalization by macrophages. Gelatin was chosen as the model material for this study, and both human- and animal-derived macrophages have been employed. The main conclusions point to a larger extent and a faster rate of internalization when using the stiffest nanoparticles.

The topic is highly appealing to the readers. As most studies focus on the influence of size, shape and/or surface functionalization of particles aimed at cell targeting and drug delivery, studies such as this one focusing on the mechanical properties of the systems have been scarce and are novel. The techniques that were used are in general sufficient to draw the conclusions claimed by the authors, but the presentation of the Results and Discussion can be significantly improved. I also found that the Introduction should be revised to provide a better motivation for this work.

In order to address the abovementioned concerns, I invite the authors to go through the points listed below:

  1. The text should be revised. The two sentences that start in lines 41 and 42 both mention "studies"; in lines 55 and 56 the function of macrophages is mentioned, but it is repeating the idea from lines 48-52. These are just a few examples that do not help with the fluency of the manuscript.

Answer: We thank the reviewer for the valuable comment and changed the above-named parts.
The first sentence mentioned is changed to the following: “Nowadays, research highlighting particle elasticity is gaining interest as intriguing biological behavior is connected.” and to avoid repetition we deleted the former lines 28 to 52: “Moreover, macrophages have an important role in tumor progression. They may show anti-tumorigenic or pro-tumorigenic activity. Furthermore, macrophages can be designed to transport drug nanoparticles (NPs) to tumor sites. Considering this role of macrophages in removal of foreign materials as well as the involvement in inflammatory processes requires the understanding of the nanoparticles’ interaction in response to varying elasticity.” 

  1. The authors introduce gelatin nanoparticles (GNPs) as biocompatible, biodegradable, recyclable, low cost and low immunogenic (lines 34-35). These properties are common to several other polymer- and lipid-based NPs, so how do GNPs differ from other drug delivery systems? The advantages are not clear.

Answer: Thank you for the comment, we extended the part about the advantages of GNPs and hope we made it more clear why gelatin is a valuable material to work with. The respective part is now: “Gelatin is a natural polypeptide obtained by acidic or alkaline hydrolysis or enzymatic degradation of collagen [1-3]. It is classified as “Generally Recognized as Safe (GRAS)” material by the United States Food and Drug Administration (FDA) [4], and it is already used in different approved pharmaceutical, dental and regenerative medicine applications including cancer therapy, tissue scaffolds and drug delivery [5]. Gelatin has a high number of functional groups and thus the resulting particles can be easily modified for targeted drug therapy [6] opening up improved applications. Gelatin nanoparticles (GNPs) are promising drug and gene carriers due to the polymers’ biocompatibility, biodegradability, recyclability, low cost, wide variety of source, good surface properties, chemical modification potential, crosslinking possibility, and low immunogenicity [7]. Besides this, the elasticity of GNPs can be adjusted [8]. The hydrophilic nature of GNPs makes it especially suitable for delivery of genetic material, peptides, and proteins [9,10].”

  1. In lines 52-55, the text focuses on cancer treatments. Since this manuscript is not directed to a specific treatment or disease, I suggest that either this example is eliminated or other examples are added to illustrate potential applications.

Answer: We thank the reviewer for the input to make the introduction more directing towards our work and removed the examples of cancer treatments.

  1. Section 2.3 can be improved. For example, how many runs were set per measurement? What temperature was selected? What type of cuvette was used, and how much volume (important since it may affect the number of counts and quality of measurement)? While they may seem trivial, this info is aligned with the level of detail and attention given to the other methods described in Section 2.

Answer: To bring the description of the size measurement to the same level as other methods described in section 2, we added cuvette type, volume, number of counts, total volume, and measurement temperature.

  1. In Section 2.4, it is mentioned that the GNPs were immobilized electrostatically (lines 137 and 138). The authors should state the charges of GNPs and polyethylenimine.

Answer: We thank the reviewer for the comment. To make it clearer, we added a sentence about the charges in the respective section. “Glutaraldehyde crosslinked GNPs are negatively charged with a surface potential of around -20 mV at neutral pH whereas bPEI coated surfaces have a positive surface charge.”

  1. In the Results and Discussion, line 229, the average size of the GNPs is given with one decimal case. Does the equipment provide an accurate reading to this number of digits? Please consider rounding up to the units. The authors should also mention if these are mean values or the upper and lower values obtained from individual experiments.

Answer: Even though the device is giving numbers with one decimal case we agree with the referee that this seems not to be too meaningful. Thus, rounded them up. Additionally, to clarify the meaning of the given number we added the following sentence: ‘’The results are expressed as mean the value of each batch’’ is added to line 229.

  1. I am also concerned about the claim provided in lines 231 and 232: "to only change the elasticity of the carrier system without influencing other colloidal properties". At this point only the results concerning the size have been shown, whereas the elasticity assessment only comes in Section 3.2: this claim could be made there, but not in Section 3.1. The authors should discuss if the size of the GNPs is adequate for the subsequent internalization studies (e.g. do NPs of this size usually enter cells easily?), and if it was expected that the size would affect the elasticity.

Answer: It is a valuable comment that we cannot claim results before we show them. Therefore, we moved lines 231 and 232 to section 3.2.       
To address the suitability of the particle size we added the following part to section 3.1: “The literature for the size dependency of cellular uptake of submicron particles is not consistent. While Leclerc et al claimed particles around 250 nm to be internalized with the lowest efficiency [45], Minchin et al. showed cellular interaction for RAW 264.7 between 90 and 1000 nm and slight decrease in particle-cell interaction for particles smaller than 250 nm for dTHP-1 cells. Therefore, the sizes of the obtained particles seem to be suitable for the subsequent cell internalization studies and no further optimization regarding the particle size is needed.”        
The sizes of both formulations are very similar. The small difference of 10 to 20 nm will neither influence the cellular uptake nor the particle elasticity.

  1. The distribution plots of the GNPs should also be shown, not just the z-average. This would show alongside the PDI how the crosslinking affects the size distribution.

Answer: We presented the particle size not only by the mean z-average together with the PdI (indicating the particle size distribution) but also within the AFM images the narrow particle size distribution is reflected. However, we added the size distribution graph, which is shown below, to the supporting information and refer to them in section 3.1

  1. The authors state in line 242 the sizes of the GNPs measured in the AFM. Again, please check the decimal cases.

Answer: Same as before, AFM is able to measure such differences but finally the value is an average and by this the digit is not relevant at all. Consequently, we adjusted the significant digits und rounded the numbers up to the unit.

  1. Later it is mentioned that the AFM was made in liquid conditions, but this was not clear from the Materials and Methods. Please revise the experimental description in Section 2.4.

We thank the reviewer for the comment. Even though it was already stated that measurements were performed in MiliQ-Water we slightly changed this section and hope it is clearer now that measurements were made in liquid.

  1. In line 249 a statistical significance is indicated (p <0.001). The p-value should be placed in the figure caption, not the Discussion.

Answer: The p-value is deleted from the former line 249, it was already placed in the caption of Fig. 3, therefore, no changes were implemented in the caption.

  1. The authors used dTHP-1 and RAW 264.7 cells to make the cell experiments, but the motivation for these choices is not clear. For example, dTHP-1 differentiates from the THP-1 human monocytic leukemia cell line. What are the benefits of choosing this specific cell line?

Answer: We added a section about both cell lines to the introduction and hope that the idea of using dTHP-1 and RAW 264.7 cells is clearer now.

  1. In line 307 differences in internalization are justified by one macrophage being human and the other from mouse. Do the authors know how they differ (e.g. is the cell surface or cell metabolism different)?

Answer: We thank the reviewer for the interesting question. The differences between THP-1 and RAW 264.7 cells are added in the introduction section. To avoid repetition, we decided not to mention them again around the former line 307.

  1. In Section 3.5, lines 342-346 are overlapping with the Introduction. Please consider moving this part to the Introduction and rewrite accordingly.

Answer: The respecting lines are deleted in the new version of the manuscript as similar content is already mentioned in the introduction in ,

Round 2

Reviewer 2 Report

The authors have significantly improved their manuscript by applying the recommended suggestions.